# Healthy core: Harmonizing brain MRI for supporting multicenter migraine classification studies

Hyunsoo Yoon[1], Todd J. Schwedt[2,3], Catherine D. Chong[2,3], Oyekanmi Olatunde[4], Teresa Wu[3,5]*

1 Department of Industrial Engineering, Yonsei University, Seoul, Republic of Korea, 2 Department of Neurology, Mayo Clinic, Scottsdale, Arizona, United States of America, 3 ASU-Mayo Center for Innovative Imaging, Tempe, Arizona, United States of America, 4 Department of Systems Science and Industrial Engineering, Binghamton University, Binghamton, New York, United States of America, 5 School of Computing and Augmented Intelligence, Arizona State University, Tempe, Arizona, United States of America

* Teresa.Wu@asu.edu

## Abstract

Multicenter and multi-scanner imaging studies may be necessary to ensure sufficiently large sample sizes for developing accurate predictive models. However, multicenter studies, incorporating varying research participant characteristics, MRI scanners, and imaging acquisition protocols, may introduce confounding factors, potentially hindering the creation of generalizable machine learning models. Models developed using one dataset may not readily apply to another, emphasizing the importance of classification model generalizability in multi-scanner and multicenter studies for producing reproducible results. This study focuses on enhancing generalizability in classifying individual migraine patients and healthy controls using brain MRI data through a data harmonization strategy. We propose identifying a 'healthy core'—a group of homogeneous healthy controls with similar characteristics—from multicenter studies. The Maximum Mean Discrepancy (MMD) in Geodesic Flow Kernel (GFK) space is employed to compare two datasets, capturing data variabilities and facilitating the identification of this 'healthy core'. Homogeneous healthy controls play a vital role in mitigating unwanted heterogeneity, enabling the development of highly accurate classification models with improved performance on new datasets. Extensive experimental results underscore the benefits of leveraging a 'healthy core'. We utilized two datasets: one comprising 120 individuals (66 with migraine and 54 healthy controls), and another comprising 76 individuals (34 with migraine and 42 healthy controls). Notably, a homogeneous dataset derived from a cohort of healthy controls yielded a significant 25% accuracy improvement for both episodic and chronic migraineurs.

## 1. Introduction

Neuroimaging studies have elucidated structural and functional alterations in brain regions involved in sensory-discriminative, affective, and cognitive pain processing, pain modulation,

**Data Availability Statement:** Data from the study sponsored by the United States Department of Defense (DOD) will be made available through the Federal Interagency Traumatic Brain Injury Research (FITBIR) Informatics System in

accordance with the rules and regulations of the DOD. Patient consent for the NIH-sponsored study and for the Mayo-funded study did not include a data sharing agreement. Researchers can request the access to the data via https://fitbir.nih.gov/content/access-data.

**Funding:** YES, This work was supported by the United States Department of Defense W81XWH-15-1-0286, Schwedt, Todd, (https://cdmrp.health.mil/search.aspx?LOG_NO=PR140037) National Institutes of Health K23NS070891, Schwedt, Todd, (https://report.nih.gov/reportweb/web/displayreport?rid=568&ver=5) and internal funds from the Mayo Clinic (Schwedt, Todd). There was no additional external funding received for this study.

**Competing interests:** The authors have declared that no competing interests exist.

and multisensory integration in patients with migraine [1–7]. Advances in machine learning have facilitated the development of models capable of distinguishing individuals with migraine from healthy controls (HCs) using data from functional magnetic resonance imaging (fMRI) and structural MRI, demonstrating their potential in migraine discrimination. However, a persistent challenge is the poor generalizability of these algorithms across different datasets, underlining the importance of model adaptability in multi-scanner and multicenter studies for ensuring reproducible results.

Harmonization has emerged as a critical endeavor in clinical research, aimed at refining, reviewing, and standardizing common data elements to improve the generalizability of classification models. This strategy, known as data curation, combines data from various sources to systematically eliminate site or cohort-specific biases, thereby enhancing the accuracy and reliability of classification models across diverse datasets. [8–11]. Such an approach specifically focuses on the harmonization of neuroimaging data in the post-acquisition phase, emphasizing the refinement and standardization of extracted features rather than adjustments during the acquisition phase, such as modifying sequence parameters.

Recent studies have highlighted the effectiveness of statistical techniques like ComBat harmonization for mitigating site-specific effects in neuroimaging data [12], and the utility of deep learning approaches in achieving robust harmonization across different datasets [13, 14]. Additionally, investigations into intensity harmonization techniques have underscored their importance in ensuring data consistency across multisite studies, notably in conditions like Parkinson's disease [15], thereby stressing the crucial role of harmonization in augmenting the reliability and comparability of neuroimaging analyses [16].

This backdrop sets the stage for addressing the inherent challenges posed by dataset heterogeneity, especially within HC cohorts, through domain adaptation. Domain adaptation leverages labeled data from one or more related source domains to build a machine learning model for unseen or unlabeled data in a target domain, aiming to discover a shared feature representation that minimizes distribution differences while preserving the original datasets' key properties [17, 18]. Despite the diversity of domain adaptation approaches, such as re-weighting, parameter adaptation, feature augmentation, and feature transformation, the need for a novel concept arises to address the overlooked heterogeneity within HC cohorts effectively.

We propose the innovative concept of the "healthy core," which entails the strategic transformation and alignment of brain MRI data from HCs, focusing on harmonizing their distinct imaging characteristics across different datasets. Unlike traditional methods, this approach specifically targets the variability among HCs from various sources to establish a consistent and representative healthy core. Our objectives are twofold: to identify and unify HCs from diverse datasets into a singular healthy core and to leverage this healthy core in the development, validation, and application of medical imaging classification models. This strategy is poised to significantly enhance the models' generalizability and accuracy, particularly for classifying migraine in novel, unseen datasets. We posit that classification models grounded in a healthy core will demonstrate superior accuracy compared to those that do not utilize this approach, thereby significantly advancing the field of migraine classification through improved model robustness and applicability.

## 2. Method

### 2.1. Approvals and consent

All studies were approved by the Mayo Clinic and Washington University School of Medicine in St. Louis Institutional Review Boards. All participants provided written consent prior to study participation. Individuals with migraine were identified from the headache clinics at

both institutions. HCs and individuals with migraine were enrolled from a database of research volunteers, and via community outreach.

## 2.2. Study participants

Diagnoses of episodic migraine (EM) and chronic migraine (CM) were assigned using the most recent version of the International Classification of Headache Disorders available at the time of enrollment [19]. Migraine diagnoses were assigned by a headache specialist (TS). Participants were recruited for the study between 2012 and 2021. All participants were male or female adults between the ages of 18–65 years old. Those with contraindications to MRI, acute pain conditions other than migraine, neurological disorders other than migraine, women who were pregnant, and individuals with abnormal brain MRI findings according to usual clinical interpretation were excluded. HCs were excluded if they had a history of migraine, however, occasional tension-type headaches ($< 3$ tension-type headaches per month) were allowed.

## 2.3. Data collection

Data collected from all study participants included age and sex. Additional data collected from migraine participants included headache frequency, number of years lived with headache, and Migraine Disability Assessment (MIDAS) scores.

Dataset 1 (DS1): The DS1 cohort, a total of 120 individuals (66 with migraine and 54 HCs) includes some patients scanned at Mayo Clinic Arizona and other patients scanned at Washington University School of Medicine in St. Louis.

Dataset 2 (DS2): The DS2 cohort, a total of 76 (34 with migraine and 42 HCs), consists of data from individuals scanned only at Mayo Clinic Arizona.

Study participants were imaged on one of two Siemens (Erlangen, Germany) scanners, each at a different institution. Scanners at both institutions differed in scanner model (MAGNETOM Trio vs MAGNETOM Skyra), use of headcoil (12-channel vs 20-channel), and T1-weighted and T2-weighted acquisition parameters, described in detail below.
Imaging Parameters:

Washington University: MAGNETOM Trio 3T scanner using a 12-channel head matrix coil.

- T1-weighted images: TE = 3.16 ms, TR = 2.4 s, 1x1x1 mm$^3$ voxels, 256x256 mm$^2$ field-of-view (FOV), acquisition matrix 256x256.

- T2-weighted images: TE = 88 ms, TR = 6280 ms, 1x1x43 mm voxels, 256x256 mm2 field-of-view, acquisition matrix 256x256.

Mayo Clinic: MAGNETOM Skyra 3T scanner using a 20-channel head matrix coil.

- T1-weighted images: TE = 3.03 ms; TR = 2.4 s; 1x1x1.25 mm3 voxels; 256x256 mm2 field-of-view, acquisition matrix 256x256.

- T2-weighted images: TE = 84 ms; TR = 6800 ms; 1x1x4 mm3 voxels; 256x256 mm2 FOV, acquisition matrix 256x256.

T-weighted images with structural abnormalities were excluded from further analyses. Some of the participant data included in this study had been included in prior publications. [20, 21]. T1-weighted image processing was performed using the automated FreeSurfer image analysis suite (version 5.3, http://surfer.nmr.mgh.harvard.edu/) available at the time of imaging. Image processing included skull stripping, automated Talairach transformation,

**Table 1. Demographics and clinical characteristics of study participants, including healthy controls (HC), episodic migraine (EM), and chronic migraine (CM) patients from two datasets (DS1 and DS2).** The table includes sample size, mean age with standard deviation, gender distribution (female/male), and the presence of aura.

| | Group | Sample Size | Age (Mean±SD) | Gender (F/M) | Aura (Yes/No) |
|---|---|---|---|---|---|
| DS1 | HC | 54 | 30.17±10.97 | 39/15 | - |
| | EM | 51 | 35.97±11.55 | 39/12 | 22/29 |
| | CM | 15 | 34.75±11.00 | 13/2 | 6/8 |
| DS2 | HC | 42 | 38.68±10.43 | 19/23 | - |
| | EM | 8 | 39.62±10.97 | 2/6 | 3/5 |
| | CM | 26 | 30.17±10.97 | 39/15 | 15/11 |

segmentation of subcortical gray and white matter, intensity normalization, gray-white mater boundary tessellation, and surface deformation [22–25]. The recon-all pipeline in FreeSurfer allows for the reconstruction of cortical surfaces. By applying the Desikan-Killiany atlas, we obtained measurements of volume, area, and thickness for cortical regions. Subcortical regions were not extracted as part of this analysis.

DS1 consisted of 120 subjects (Age: 36 (mean) ± 11 (standard deviation) years old, female: 91, male: 29, Amongst those with migraine, an average headache frequency was: 8.91 ± 6.25 days per month), including 54 healthy control (HC), 51 episodic migraine (EM) patients, and 15 chronic migraine (CM) patients. DS2 consisted of 76 subjects (Age: 38 ± 10 years old, female: 41, male: 35, the Average headache frequency for those with migraine: 17.97 ± 8.08 days per month), including 42 HCs, 8 EM patients, and 26 CM patients. The overall sample numbers and the proportion of individuals with aura in this study are shown in Table 1.

## 2.4. Evaluation of dataset differences between DS1 and DS2

First, the mean of area, thickness, and volume features are significantly different between the HCs in the two datasets with a $p$-value<0.01 before the False Discovery Rate (FDR) correction. Further, three imaging features among those 41 significant features were identified as significantly different between two cohorts by the two-sample t-test and Kolmogorov-Smirnov test after FDR correction. Fig 1 illustrates the significant mean difference in each imaging feature between the two cohorts. This result confirms that there are mean differences between the two cohorts for each modality (mean, thickness, and volume features).

Second, the variances of the features were tested. Table 2 shows that the variance for each feature is not significantly different between two cohorts under both tests after FDR correction.

Furthermore, we investigated the possibility that significant group differences in sex distribution ($p$-value = 0.01) contributed to the cohort difference. We randomly selected the sex-balanced sub samples from both cohorts. After making the balanced groups, the mean and variance of features (volume, area, and thickness) were tested using the t-test and the Kolmogorov-Smirnov (KS) test for mean, F-test and Ansari-Bradley test for variance whether the dataset difference exists. Since we selected a random subset to test, this test procedure was repeated 10,000 times to avoid a sampling-dependent conclusion. It was concluded that there is a significant mean difference and a marginal variance difference in each feature (volume, area, and thickness), but sex distribution was not a significant factor in the dataset difference.

Table 3 provides an overview of the average differences in means across various scenarios, including DS1 vs DS1, DS2 vs DS2, random splits, and DS1 vs DS2. In the case of DS1 vs DS1, we examined 21 samples randomly selected from the DS1 dataset and another 21 non-

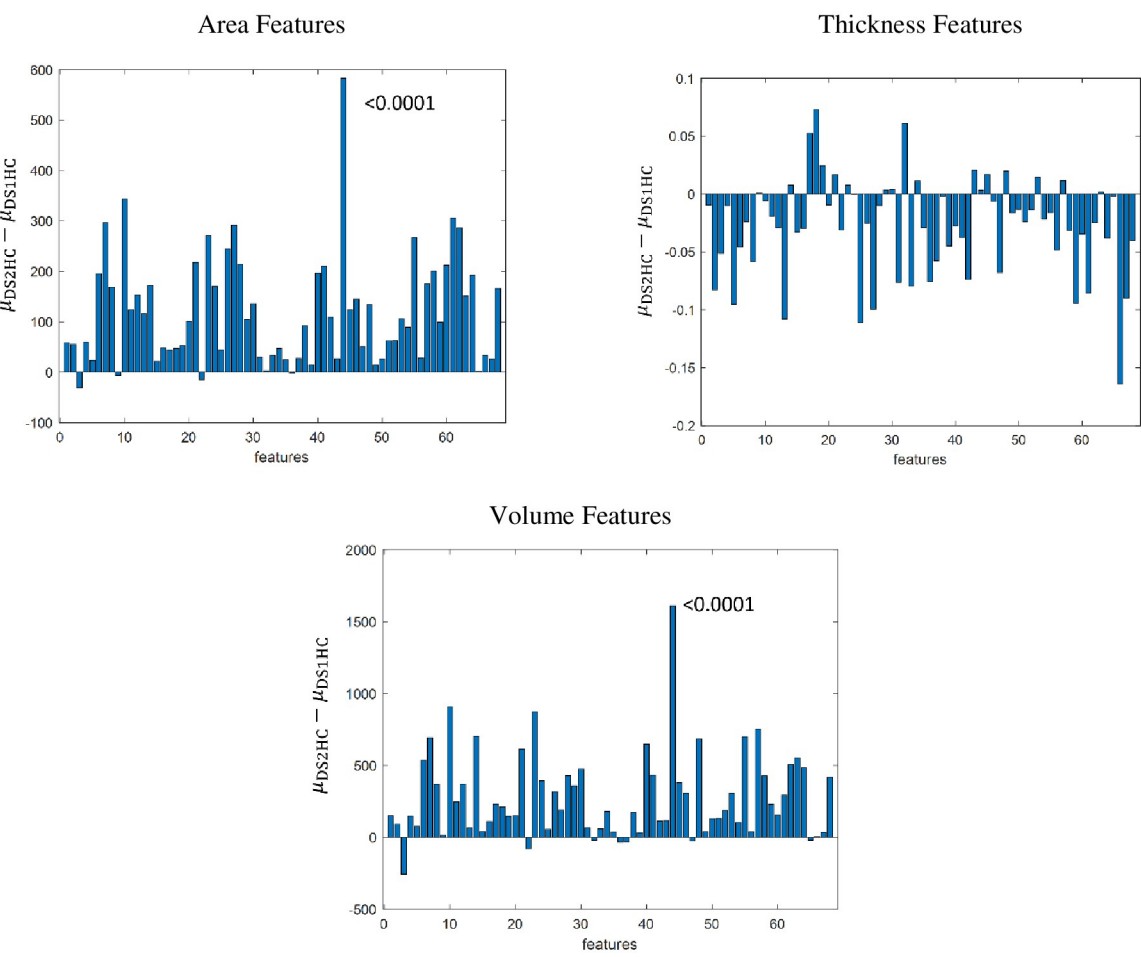

**Fig 1. Univariate analysis for mean difference scale.** Note: $\mu_{\text{DS1HC}}$ represents the Feature Average of Healthy controls in Dataset 2 and $\mu_{\text{DS1HC}}$ depicts the Feature Average of Healthy controls in Dataset 1.

overlapping samples from the same dataset. It is important to note that in Dataset 2, there are a total of 42 Healthy controls, which was evenly divided between the two datasets. We ensured that each subgroup had a maximum of 21 samples, chosen carefully for consistency and balance across all four scenarios. For the 'random splits' scenario, we randomly select 21 subsets from the combined dataset (DS1 and DS2) and included another 21 non-overlapping samples. The numbers presented in the table represent the mean features for each subset and the variations between them. Given the inherent randomness in the sampling process, we repeated this procedure 100 times and reported the averaged results.

Table 3 confirms there is more of a difference between the two datasets (DS1 vs DS2) compared to sample subsets from within a dataset (e.g., DS1 vs DS1). Building on the comparative

**Table 2. The difference in the variances of features between DS1 and DS2.**

|  | Two-sample F-test for equal variances | Ansari-Bradley Test for Equal Variances |
|---|---|---|
| Before FDR | 8 | 9 |
| After FDR | 0 | 0 |

**Table 3. Average mean difference under random subsets within dataset.**

| Avg. \|DS1-DS2\| | DS1 vs DS1 | DS2 vs DS2 | Random splits | DS1 vs DS2 |
|---|---|---|---|---|
| Area (mm$^2$) | 82.12 | 77.86 | 80.75 | 102.27 |
| Thickness (mm) | 0.0416 | 0.0458 | 0.0436 | 0.0481 |
| Volume (mm$^3$) | 253.74 | 257.07 | 256.01 | 275.58 |

analysis presented in Table 3, we further quantified the average mean differences after mean and variance adjustments, as shown in Table 4. This additional analysis offers deeper insight into the impact of these adjustments on the dataset characteristics. The discrepancies illustrated in Table 4 validate our approach by highlighting the persistent differences between DS1 and DS2, even after standard mean and variance adjustments. Simple cohort-wise adjustments still have limited clinical applicability. This finding motivated us to develop new methods to construct a "healthy core" that consists of HC with similar characteristics from DS1 and DS2.

To ensure the robustness and clinical applicability of our classification models, hyperparameters were optimized through Bayesian optimization, targeting the maximization of the Area Under the Receiver Operating Characteristic Curve (AUC) on the validation dataset. This decision to prioritize AUC as our optimization criterion stems from its comprehensive ability to balance sensitivity and specificity—critical for the intended clinical use of our models.

## 2.5. Constructing the healthy core

The heterogeneity inherent in multicenter studies, arising from variations in imaging protocols, scanner types, and participant populations, necessitates a sophisticated approach to harmonize control groups. Our strategy to construct a homogenous group of Healthy Controls (HCs) is twofold: first, we assess the overall similarity between HCs from different datasets using a collective analysis of multiple MRI features; second, we evaluate the data's dimensional representation for a deeper similarity assessment. Central to our method is the application of Maximum Mean Discrepancy (MMD) [26–31], a robust kernel-based statistic that evaluates the similarity between distributions of features from different datasets. Unlike traditional applications that might assess features individually, our approach employs MMD across the comprehensive feature vector, encompassing all 204 MRI-derived measures collectively. This holistic application ensures a nuanced and thorough determination of homogeneity across datasets, acknowledging the multidimensional nature of brain imaging data. To address the potential rigidity of MMD due to its point estimate nature and to accommodate the inherent variability between datasets, we integrate the Geodesic Flow Kernel (GFK). GFK, which operates on the Grassmannian manifold, offers a dynamic analysis by constructing a continuous transformation between the feature spaces of the two datasets. This flexible and accurate identification of homogenous HCs. The synergy between MMD and GFK in our framework facilitates a robust identification of a 'Healthy Core'—a subset of HCs whose features are consistently representative across different studies. This 'Healthy Core' construction process is visualized in Fig 2, where we depict how the combined use of MMD and GFK underpins our

**Table 4. Average mean difference after mean and variance adjustments under random subsets within dataset.**

| Avg. \|DS1-DS2\| | DS1 vs DS1 | DS2 vs DS2 | Random splits | DS1 vs DS2 |
|---|---|---|---|---|
| Area (mm$^2$) | 0.2488 | 0.2387 | 0.2485 | 0.3262 |
| Thickness (mm) | 0.2479 | 0.2484 | 0.2432 | 0.2679 |
| Volume (mm$^3$) | 0.2288 | 0.2509 | 0.2491 | 0.2700 |

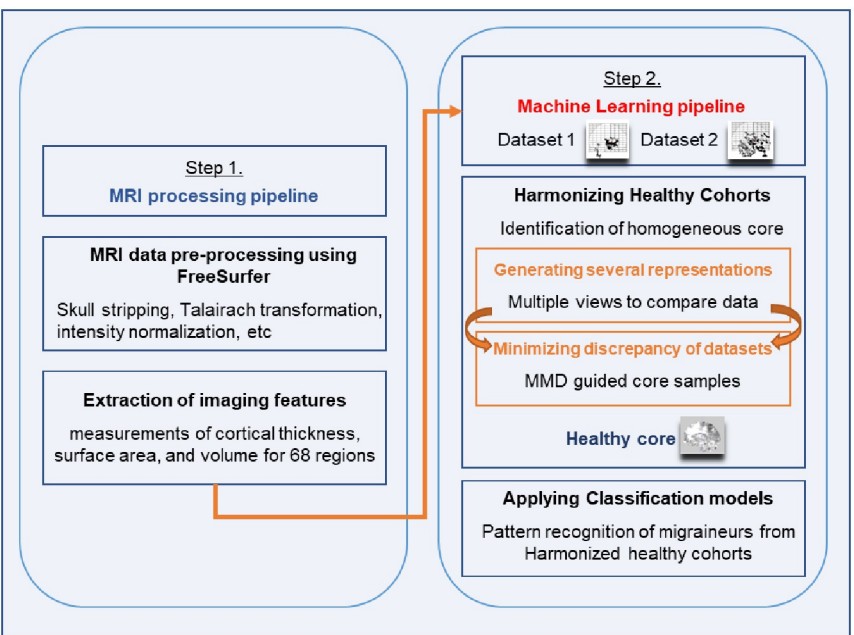

**Fig 2. Workflow for constructing the healthy core using the 'multiple view' strategy and MMD: This illustration outlines our approach to harmonize a homogenous healthy control group using 'multiple view' analysis, which examines MRI features from diverse analytical perspectives.** Crucially, we apply the Maximum Mean Discrepancy (MMD) across multiple features, not just individually, leveraging Geodesic Flow Kernel (GFK) for comprehensive dataset evaluation. This ensures the selection of a consistent healthy core, enhancing the robustness and generalizability of our classification models across multicenter studies.

novel design strategy. By employing these advanced statistical tools, we not only mitigate the cohort effects commonly encountered in multicenter studies but also enhance the accuracy and generalizability of our migraine classification models. A detailed description of our unified GFK+MMD framework and its implementation is provided in the S1 Appendix.

We hypothesize that using a healthy core, a homogenous subset of HCs from different sources, can help improve the generalization of the machine learning model for classifying migraine since it addresses the heterogeneity of the HCs within and across the datasets; and sets the bridges between the datasets. To comprehensively assess the clinical utility of the proposed method, we designed three sets of experiments: (1) evaluating the performance of migraine classification models within a single dataset; (2) evaluating the performance of migraine classification models developed using one dataset and tested on a second dataset; and (3) evaluating the performance of migraine classification models with and without using healthy core samples for classifying individuals with CM and those with EM from a different dataset. In the training of the classification models, cross-validation has been applied. To get reliable results for the relatively small sample sizes, experiments are repeated under five different random scenarios (cross-validated) to build classifiers and report average performances. Since our last experiment focuses on migraine only, we report the Area Under the Curve (AUC) for the first two experiments and the accuracy of identifying migraine in the last experiment.

## 3. Results

In our investigation, we constructed four binary classifiers: an $L_1$ regularized logistic regression model, support vector machine (SVM), random forest, and XGBoost. These models were

**Table 5. Data split design for Experiments I, II, III overview of training, validation, and test datasets used in three experiments.** Each row indicates the specific experiment, dataset, and the comparison group, healthy controls (HC), episodic migraine (EM), and chronic migraine (CM) patients.

| Experiment | Dataset | Training | Validation | Test |
|---|---|---|---|---|
| I | DS1 | HC vs. CM | HC vs. CM | HC vs. CM |
| | DS2 | HC vs. EM | HC vs. EM | HC vs. EM |
| II | DS1 → DS2 | HC vs. CM | HC vs. CM | HC vs. CM |
| | DS2 → DS1 | HC vs. EM | HC vs. EM | HC vs. EM |
| III | DS1 → DS2 | HC Core vs. CM | HC Core vs. CM | HC Core vs. CM |
| | DS2 → DS1 | HC Core vs. EM | HC Core vs. EM | HC Core vs. EM |

tasked with distinguishing between Healthy Controls (HC) and Chronic Migraineurs (CM), as well as HC and Episodic Migraineurs (EM). A central aspect of our study was the emphasis on the clinical applicability of the 'healthy core,' derived from original data to avoid the potential pitfalls associated with artificial data generation methods. Notably, our utilization of random forest and XGBoost models inherently benefits from resampling techniques like bootstrapping and bagging, integral to their design. This methodological choice demonstrates our commitment to enhancing the generalization of our results while ensuring the preservation of data integrity, thus highlighting the 'healthy core's' utility in clinical contexts without compromising the authenticity of the utilized data. The experiments conducted in this study utilized distinct training, validation, and test datasets to evaluate the performance of the classification models. Table 5 provides an overview of these datasets and their usage across three different experiments.

### 3.1. Experiment I: Evaluating the performance of different classification models within a single dataset

The objective of Experiment I was to evaluate the effectiveness of our machine learning models in accurately classifying migraine conditions within individual datasets, as detailed in Table 5. We applied the four models—$L_1$ regularized logistic regression, SVM, random forest, and XGBoost—to each dataset (DS1 and DS2), yielding the following outcomes:

The average AUC of the four models for HC vs CM was 0.7533 for the DS1 test dataset and 0.6465 for the DS2 test dataset as seen in Table 6A. The average AUC of the four models for HC vs EM was 0.6965 for the DS1 test dataset and 0.6135 for the DS2 test dataset as seen in Table 6B. We conclude that the machine learning models provide good performance during training, but that the performance deteriorates when using test data. This deterioration in accuracy is expected.

### 3.2. Experiment II: Evaluating the performance of the classification models developed on one dataset and tested on a second dataset

The goal is to assess the generalizability of machine learning models across datasets as shown in Table 7. Similar to Experiment I, we tested on HC vs. EM and HC vs. CM. In this experiment, one dataset was fully utilized for training and validation, while testing was conducted on the second dataset. The outcomes are detailed as follows:

As seen in Table 7A for HC vs CM, all four trained classifiers did not perform well in this cross-dataset experiment. The average AUC on models trained on DS1 dropped from 0.7533 (testing on DS1) to 0.5245 (testing on DS2), and the average AUC on models trained on DS2 dropped from 0.6465 (testing on DS2) to 0.6230 (testing on DS1). We can also observe similar patterns for HC vs EM in the cross-dataset experiment as seen in Table 7B. The average AUC

**Table 6.**

| | A. AUC of Four Classification Models in Single Dataset Experiments for CM | | | |
|---|---|---|---|---|
| | Training on DS1 (Validation) | Test on DS1 (Test) | Training on DS2 (Validation) | Test on DS2 (Test) |
| Regularized Logistic Reg. | 0.9600 | 0.7360 | 0.9820 | 0.6380 |
| SVM | 0.8720 | 0.6700 | 0.9800 | 0.5900 |
| Random Forest | 0.9320 | 0.7700 | 0.9180 | 0.6560 |
| XGBoost | 0.9480 | 0.8370 | 0.9040 | 0.7020 |
| Average | | 0.7533 | | 0.6465 |
| | B. AUC of Four Classification Models in Single Dataset Experiments for EM | | | |
| | Training on DS1 (Validation) | Test on DS1 (Test) | Training on DS2 (Validation) | Test on DS2 (Test) |
| Regularized Logistic Reg. | 0.8960 | 0.6660 | 0.8920 | 0.5760 |
| SVM | 0.9480 | 0.6740 | 0.9500 | 0.6060 |
| Random Forest | 0.9960 | 0.7400 | 0.9420 | 0.6280 |
| XGBoost | 0.8440 | 0.7060 | 0.9320 | 0.6440 |
| Average | | 0.6965 | | 0.6135 |

on models trained on DS1 dropped from 0.6965 (testing on DS1) to 0.5781 (testing on DS2), and the average AUC on models trained on DS2 dropped from 0.6135 (testing on DS2) to 0.5963 (testing on DS1). This result demonstrates the existence of dataset discrepancies that need to be cautiously handled when building classification models.

## 3.3. Experiment III: Evaluating the performance of classification models with and without using healthy core

The goal of Experiment III was to assess the applicability of machine learning models developed in one dataset, including participants with migraine and the healthy core, to classify individuals with migraine from a separate dataset. The experimental setup for cross-dataset experiments with and without the Healthy Core are shown in Table 5.

The implementation of the healthy core concept markedly enhances the performance and generalizability of our classification models. Prior to integrating the healthy core, the models

**Table 7.**

| | A. AUC of Four Classification Models for HC vs CM in Cross-Dataset Experiments | | | |
|---|---|---|---|---|
| | Training (DS1) (Validation) | Test (DS2) (Test) | Training (DS2) (Validation) | Test (DS1) (Test) |
| Regularized Logistic Reg. | 0.9160 | 0.5040 | 0.9340 | 0.6160 |
| SVM | 0.9740 | 0.580 | 0.8620 | 0.5980 |
| Random Forest | 0.9640 | 0.4400 | 0.9820 | 0.6900 |
| XGBoost | 0.9940 | 0.5740 | 0.8680 | 0.5880 |
| Average | | 0.5245 | | 0.6230 |
| | B. AUC of Four Classification Models for HC vs EM in Cross-Dataset Experiments | | | |
| | Training (DS1) (Validation) | Test (DS2) (Test) | Training (DS2) (Validation) | Test (DS1) (Test) |
| Regularized Logistic Reg. | 0.8475 | 0.6375 | 0.9100 | 0.5900 |
| SVM | 0.7850 | 0.6050 | 0.9725 | 0.5825 |
| Random Forest | 0.9600 | 0.5325 | 0.9675 | 0.5950 |
| XGBoost | 0.8225 | 0.5375 | 0.9075 | 0.6175 |
| Average | | 0.5781 | | 0.5963 |

**Table 8. Accuracy, specificity, sensitivity, and F1-score of four models with and without healthy core for CM and EM classification.**

| | | Regularized LR | | SVM | | Random Forest | | XGBoost | |
|---|---|---|---|---|---|---|---|---|---|
| | | Without Healthy Core | With Healthy Core | Without Healthy Core | With Healthy Core | Without Healthy Core | With Healthy Core | Without Healthy Core | With Healthy Core |
| Predicting CM from DS2 | Accuracy | 0.4900 | 0.6780 | 0.3790 | 0.7833 | 0.4700 | 0.7680 | 0.5650 | **0.8740** |
| | Specificity | 0.5597 | 0.7322 | 0.4329 | 0.8460 | 0.5369 | 0.8294 | 0.6454 | **0.9439** |
| | Sensitivity | 0.4253 | 0.6276 | 0.3289 | 0.7251 | 0.4079 | 0.7109 | 0.4903 | **0.8091** |
| | F1-score | 0.5138 | 0.6865 | 0.4017 | 0.7899 | 0.4938 | 0.7749 | 0.5883 | **0.8783** |
| Predicting CM from DS1 | Accuracy | 0.6100 | 0.6920 | 0.4175 | 0.6325 | 0.4800 | 0.8325 | 0.5840 | **0.8400** |
| | Specificity | 0.7571 | 0.6712 | 0.5181 | 0.6135 | 0.5957 | 0.8075 | 0.7248 | **0.8148** |
| | Sensitivity | 0.3355 | 0.7383 | 0.2296 | 0.6748 | 0.2640 | 0.8882 | 0.3212 | **0.8962** |
| | F1-score | 0.3751 | 0.6216 | 0.2157 | 0.5526 | 0.2616 | 0.8010 | 0.3501 | **0.8113** |
| Predicting EM from DS2 | Accuracy | 0.4716 | 0.7510 | 0.5550 | **0.8787** | 0.5950 | 0.8520 | 0.5520 | 0.8250 |
| | Specificity | 0.4575 | 0.7285 | 0.5383 | **0.8523** | 0.5771 | 0.9800 | 1.0000 | 1.0000 |
| | Sensitivity | 0.4756 | 0.7574 | 0.5598 | **0.8862** | 0.6001 | 0.8154 | 0.4240 | 0.7750 |
| | F1-score | 0.2779 | 0.6653 | 0.3497 | **0.7575** | 0.3878 | 0.7464 | 0.4980 | 0.7175 |
| Predicting EM from DS1 | Accuracy | 0.4180 | 0.6250 | 0.4950 | 0.7400 | 0.5560 | 0.6600 | 0.4920 | **0.7920** |
| | Specificity | 0.4647 | 0.7750 | 0.4752 | 0.7911 | 0.5338 | 0.7098 | 0.4723 | **0.8424** |
| | Sensitivity | 0.5587 | 0.7375 | 0.4848 | 0.7323 | 0.4262 | 0.6326 | 0.4877 | **0.7043** |
| | F1-score | 0.5445 | 0.7253 | 0.5406 | 0.7916 | 0.5774 | 0.7076 | 0.5387 | **0.8422** |

exhibited limited ability to generalize across datasets, with test accuracies for classifying Chronic Migraine (CM) at 0.4760 for DS2 and 0.5229 for DS1, and for Episodic Migraine (EM) at 0.5434 for DS2 and 0.4903 for DS1. The incorporation of the healthy core into the model development process, however, led to substantial improvements across all key metrics —Accuracy, Specificity, Sensitivity, and F1-Score—demonstrated in Table 8. For example, the use of the healthy core improved the Accuracy for predicting CM from DS2 using XGBoost from 0.5650 to an impressive 0.8740 and similarly enhanced the Specificity and Sensitivity metrics, indicating a more balanced and precise classification capability. Similarly, for predicting EM from DS2, the Accuracy improved from 0.5950 to 0.8520 with Random Forest, demonstrating the healthy core's role in bolstering model adaptability and reliability across distinct datasets.

### 3.4. In-depth analysis of healthy core

In Section 3.1, we conducted statistical analysis to confirm the dataset discrepancy between DS1 and DS2. For illustration, here we adopted t-Distributed Stochastic Neighbor Embedding (t-SNE), a technique for dimensionality reduction (e.g., based on principal component analysis) that is particularly well-suited for the visualization of high-dimensional datasets to demonstrate the impact of a healthy core. For details of t-SNE, interested readers are referred to [32].

As seen in Fig 3A and 3C, combined HCs do not have separating (thus predictive) pattern compared to CM from either DS1 or DS2. On the other hand, the selected HCs cores have representative similar patterns, and healthy core (see Fig 3B and 3D) can be well clustered in the t-SNE plot.

Similar trends were observed for HC vs EM. Fig 4A demonstrated that the combined HCs contain mixed patterns with noise. However, 28 healthy cores identified by the proposed methods show compact representative patterns and are well separable from EM.

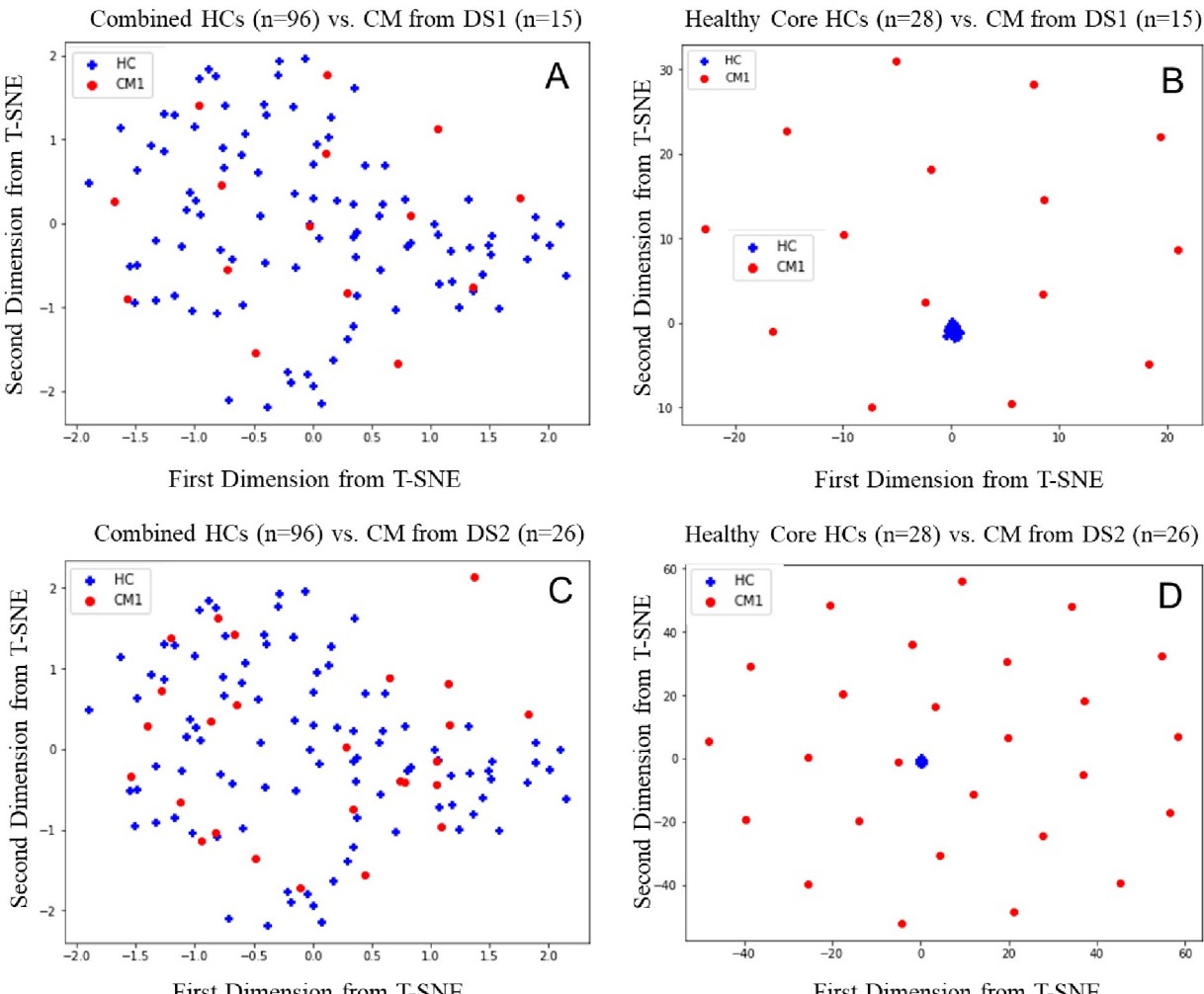

**Fig 3. t-SNE plot of DS 1 and DS2 for CM on the first and second principal component.** (A) combined HCs (n = 96) vs. CM from DS1 (n = 15). (B) Healthy Core (n = 28) vs. CM from DS1 (n = 15). (C) combined HCs (n = 96) vs. CM from DS2 (n = 26). (D) Healthy Core (n = 28) vs. CM from DS2 (n = 26).

## 4. Discussion

The inherent differences in data characteristics stem from various factors, including variations in patient demographics and differences in data acquisition methods from multiple institutions or multi-center. The key discovery of this study lies in the effective utilization of a 'healthy core,' representing a uniform dataset sourced from a group of healthy individuals. This strategy significantly improves the performance of classification models for both EM and CM. Implementing a healthy core successfully counters the usual decline in model performance when applying the migraine classification model to distinct datasets gathered from patients across diverse institutions and scanned using varying devices. Depending on the modeling approach, integrating a healthy core led to classification accuracy as high as 88% for both EM and CM.

The use of independent training and testing datasets is a substantial strength of the studies reported herein. Many prior publications (including some of our own) reported the accuracy of classification models for migraine train and test using the same dataset, with strategies such

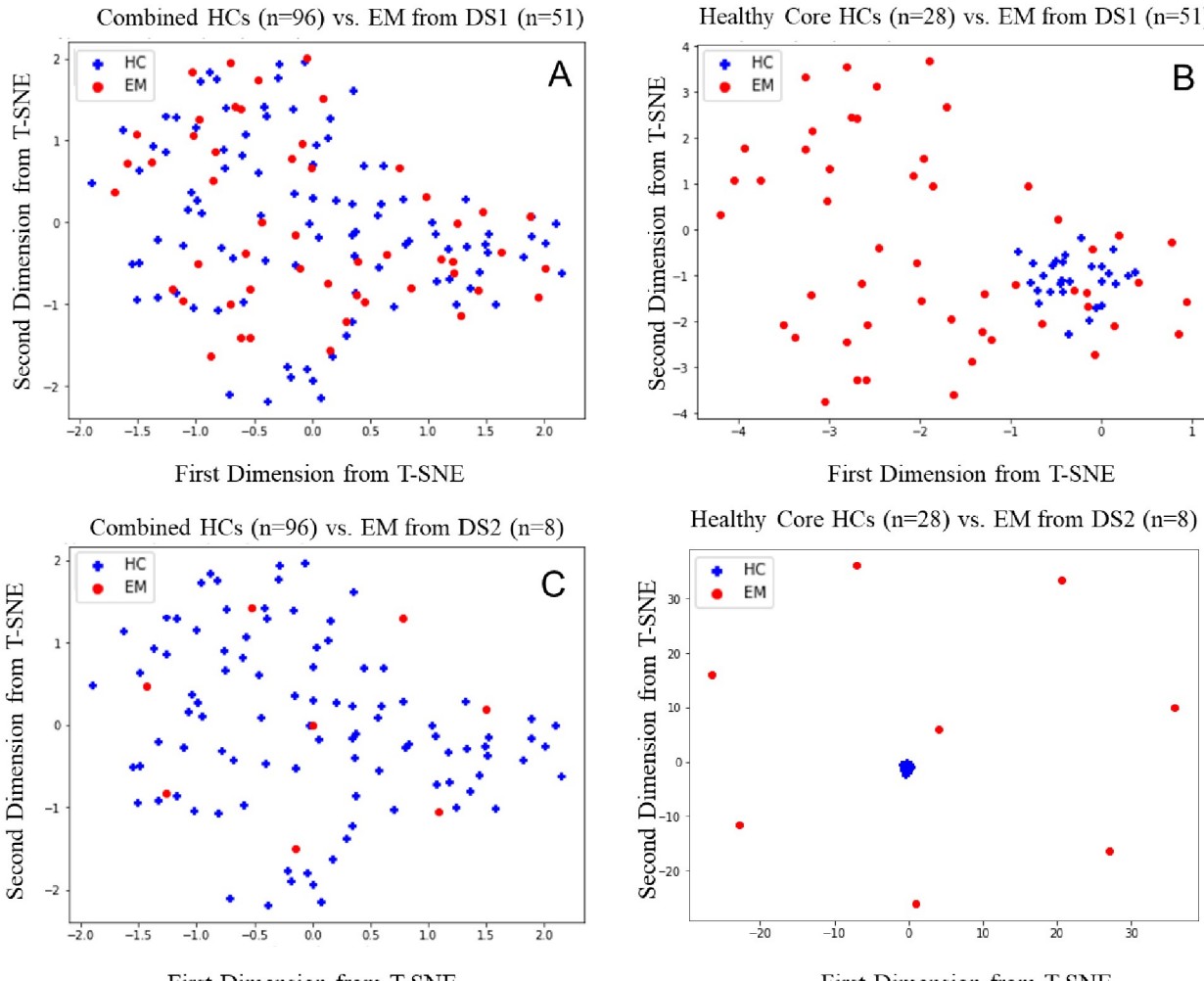

**Fig 4. t-SNE plot of DS 1 and DS2 for EM on the first and second principal component.** (A) combined HCs (n = 96) vs. EM from DS1 (n = 51). (B) healthy core (n = 28) vs. EM from DS1 (n = 51). (C) combined HCs (n = 96) vs. EM from DS2 (n = 8). (D) healthy core (n = 28) vs. EM from DS2 (n = 8).

as the "leave-one-out" methodology [20–22, 33–36]. Although this type of approach is valid and well-accepted, and therefore failing when it would be used in a completely, it likely leads to overestimating the performance of the model if it was used in a completely new and previously unseen dataset. Given that separate training and testing datasets were employed in the conducted experiments, it is advisable to view the reported classification accuracies as more cautious approximations of the model's performance.

Useful classification models need to have high performance when tested on data that are collected from new patient populations and using collection techniques that might differ slightly from those used to collect the data that were included for model training (i.e. cross dataset accuracy). For example, a brain MRI-based classification model should still have high performance when tested on data from patients imaged at a different medical center and using different MR scanners. The experiments reported in this manuscript purposefully introduce this type of heterogeneity, with participants being enrolled from two different medical centers in two different regions of the United States. Results demonstrate the expected reduction in

accuracy of the migraine classification models when they have been tested on independent datasets that include heterogeneity typical of multicenter imaging experiments. The experiments without the healthy core that are reported herein demonstrate this: CM and EM classification accuracy was higher in the single dataset experiments (CM accuracy 65%-75%, EM accuracy 61%-70%) compared to the cross-dataset experiments (CM accuracy 57%-59%, EM accuracy 58%-60%). The introduction of a healthy core helped to overcome this deteriorating model performance and provided much higher classification accuracies for CM and EM.

When performing research that includes "healthy controls" there is an assumption that the healthy control cohort is homogeneous. However, even when stringent eligibility criteria are applied in an attempt to make the healthy control group as healthy as possible, there will always be identifiable and unidentifiable heterogeneity within the healthy control group due to differences in demographics, prior life experiences, physical and mental well-being, underlying genetic heterogeneity, the existence of diseases that have not yet manifest, and other sources of heterogeneity. When establishing "normal values" for certain types of tests, such as blood test results, for example, the use of very large sample sizes can mostly overcome this issue of heterogeneity within the healthy control cohort. However, available sample sizes are smaller for establishing normal values for brain MRI, a diagnostic test that is time and cost-intensive. The "healthy core" method described in this manuscript can help overcome this challenge. The code for the proposed healthy core algorithm is readily available and can be shared upon request. For example, with the use of the healthy core and XGBoost, the cross dataset classification accuracy improved to 84%-87% for CM and to 79%-83% for EM.

These significant enhancements affirm the healthy core's utility in fostering the generalization of classification models. By leveraging a harmonized subset of HCs from different datasets, our approach not only improves model accuracy but also ensures the models are robust and applicable across varying clinical and research settings. Thus, the healthy core concept represents a significant advancement in the development of machine learning classifiers for migraine detection, as evidenced by the substantial improvements across all evaluated performance metrics.

## 5. Conclusion

When using the healthy core, the classification accuracies for migraine are comparable, if not somewhat higher, than those reported previously in the literature. This is so even though independent training and testing sets were used. For example, prior studies using brain imaging structural data reported classification accuracies of 67%-86% for differentiating migraine from healthy controls [20], while those using functional MRI data reported accuracies of 73%-86% [21, 33, 35]. Classification models including multimodality imaging data, combining structural and functional measures, have provided accuracies of 83%-84 [34, 37]. Classification accuracies for EM were generally lower than for CM in the experiments reported herein. This is consistent with expectations since CM is the more severe disease state. Prior brain imaging studies have demonstrated that headache frequency has a positive correlation with the amplitude of structural and functional brain changes, and prior classification models have had similar findings [4, 20, 37–42].

This study highlights the significant potential of a healthy core in improving the accuracy and robustness of migraine classification models. The findings support the practicality of utilizing a healthy core as a valuable tool in handling variations in imaging data across different institutions and scanners, ultimately contributing to more reliable and generalizable classification models for migraine. In conclusion, the utilization of a healthy core can increase the accuracy and generalizability of brain imaging-based classification models for EM and CM.

Inclusion of a healthy core addresses intrinsic heterogeneity that exists within a healthy control cohort and in multicenter studies, even when stringent participant eligibility criteria and methodology are used.

## 6. Limitations

We assumed heterogeneity amongst the healthy control cohort, and this assumption served as justification for developing a healthy core. However, a limitation of this study is that we are only partially able to describe the heterogeneity using the non-imaging data that were collected, such as the medical center from which a participant was enrolled, their sex, and age. Future studies should more deeply phenotype the healthy control subjects in search of sources of heterogeneity that might associate with differences in brain structure. Also, it should be noted that imaging data used in the analyses reported herein were also used in prior research on classification models for migraine [20, 37]. Future studies will continue to validate the migraine classification models reported herein and test the value of using a healthy core when developing new classification models.

Sample Size for Scalability: The study's statistical methods, MMD rely on sample sizes, denoted as 'm' and 'n,' from the distributions being compared. The quadratic dependence of MMD on sample size can pose computational challenges, particularly when dealing with substantial data volumes. While an adapted version of MMD with reduced computational cost exists, it is essential to recognize that in clinical studies with finite and manageable sample sizes, this limitation is less pronounced.

Interpretability of GFK Feature Space: The introduction of a new feature space by GFK that captures the gradual change from one dataset to another can be statistically meaningful. However, this feature space may not be directly interpretable in the context of the original data. Researchers should be aware that while GFK effectively identifies homogeneous characteristics, it is not designed for pinpointing specific differences between datasets within this novel feature space.

Data Quality: Both MMD and GFK are reliant on the quality of the data. Issues such as missing data, noise, and errors can impact the performance of these methods. Furthermore, data quality issues can affect the overall quality of the derived "healthy core." It is essential for researchers to ensure data quality and preprocessing measures to minimize the impact of these limitations.

## Supporting information

**S1 Appendix. [43].**
(DOCX)

**S1 Checklist. STROBE statement—Checklist of items that should be included in reports of observational studies.**
(DOCX)

## Author Contributions

**Conceptualization:** Hyunsoo Yoon, Todd J. Schwedt.

**Data curation:** Catherine D. Chong, Oyekanmi Olatunde.

**Formal analysis:** Todd J. Schwedt, Catherine D. Chong, Oyekanmi Olatunde, Teresa Wu.

**Funding acquisition:** Todd J. Schwedt.

**Methodology:** Hyunsoo Yoon, Teresa Wu.

**Project administration:** Todd J. Schwedt, Teresa Wu.

**Supervision:** Teresa Wu.

**Validation:** Hyunsoo Yoon, Todd J. Schwedt, Catherine D. Chong, Oyekanmi Olatunde.

**Visualization:** Oyekanmi Olatunde.

**Writing – original draft:** Hyunsoo Yoon, Todd J. Schwedt, Catherine D. Chong, Teresa Wu.

**Writing – review & editing:** Hyunsoo Yoon, Todd J. Schwedt, Catherine D. Chong, Teresa Wu.

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
