## [Decision Letter · Decision Letter 0]

19 Sep 2023

PONE-D-23-16414Harmonizing Healthy Cohorts to Support Multicenter Studies on Migraine Classification using Brain MRI DataPLOS ONE

Dear Dr. Wu,

Thank you for submitting your manuscript to PLOS ONE. After careful consideration, we feel that it has merit but does not fully meet PLOS ONE’s publication criteria as it currently stands. Therefore, we invite you to submit a revised version of the manuscript that addresses the points raised during the review process.

We look forward to receiving your revised manuscript.

Kind regards,

Lorenzo Faggioni, M.D., Ph.D.

Academic Editor

PLOS ONE

Journal Requirements:

"YES, 

This work was supported by the United States Department of Defense W81XWH-15-1-0286, Schwedt, Todd,  (https://cdmrp.health.mil/search.aspx?LOG_NO=PR140037)

 National Institutes of Health K23NS070891, Schwedt, Todd,  (https://report.nih.gov/reportweb/web/displayreport?rid=568&ver=5)

and internal funds from the Mayo Clinic (Schwedt, Todd)."

Reviewers' comments:

Reviewer's Responses to Questions

**Comments to the Author**

1. Is the manuscript technically sound, and do the data support the conclusions?

Reviewer #1: Yes

Reviewer #2: Partly

2. Has the statistical analysis been performed appropriately and rigorously? 

Reviewer #1: Yes

Reviewer #2: Yes

3. Have the authors made all data underlying the findings in their manuscript fully available?

Reviewer #1: Yes

Reviewer #2: No

4. Is the manuscript presented in an intelligible fashion and written in standard English?

Reviewer #1: Yes

Reviewer #2: Yes

5. Review Comments to the Author

Reviewer #1: This research study suggests a strategy designed to enhance the accuracy and generalizability of machine learning models applied in multicenter imaging studies. The ultimate objective is to classify individual migraine patients and healthy controls using brain MRI data. Authors employed the Maximum Mean Discrepancy (MMD) in Geodesic Flow Kernel (GFK) space, to harmonize data from two sources. Their approach consisted of identifying a group of homogeneously healthy controls, called the "healthy core". The authors suggest this methodology effectively mitigated typical confounding factors inherent in multicenter studies such as participant characteristics variations, MRI scanner disparities, and divergent imaging acquisition protocols. The study utilized two datasets: one with 120 individuals (comprising 66 migraine patients and 54 controls) and another with 76 (consisting of 34 migraine patients and 42 controls). The implementation of the "healthy core" concept reportedly led to a 25% enhancement in the accuracy of the classification models for both episodic and chronic migraine patients. The authors have proficiently conveyed their findings and offered comprehensive information about their experiments.

The primary aim of my comments (both major and minor) is to help the authors advance their study to the next level. This is achievable by addressing currently ambiguous aspects, thus enhancing the study's scientific merit and improving its readability and flow.

Please check the attachment for details

Reviewer #2: Dear authors,

Congratulations on your interesting work, which aims to tackle the challenge of having generalizable neuroimaging classification models and end up proposition a "healthy core" in the lines of "normative models", being taken by other researchers.

I believe that in the current form, the manuscript is not delivering its suitable potential, not only in clinical or physiological interpretability (the most important) and also regarding reproducibility of your work or methods by others. The materials and methods section feels incomplete or unclear, and some snipets of text seem to be misplaced in the results section. Also regarding results, be aware of the interpretation of machine-learning results, and notably the influence of having imbalanced datasets and the interpretation you derive from ROC AUC values. Maybe consider having a stronger support of someone more seasoned on those topics.

Also, I'm sending you in the attachment the pdf manuscript with suggestions of edits, comments and questions.

Please address these issues and provide a more clinical interpretation of the features chosen by the algorithms and models derived from such, to elevate your paper.

Consequently, I feel that your paper needs a major revision before being ready for publication and I'll be available to help with the review of the improved version.

Kind regards

6. PLOS authors have the option to publish the peer review history of their article (what does this mean?). If published, this will include your full peer review and any attached files.

Reviewer #1: No

Reviewer #2: **Yes: **Hugo Alexandre Ferreira

---

## [Author Response · Author response to Decision Letter 0]

14 Dec 2023

We have uploaded our response letter including point-to-point responses to the comments.

---

## [Decision Letter · Decision Letter 1]

16 Feb 2024

PONE-D-23-16414R1HealthyCore: Harmonizing Brain MRI for Supporting Multicenter Migraine Classification StudiesPLOS ONE

Dear Dr. Wu,

Thank you for submitting your manuscript to PLOS ONE. After careful consideration, we feel that it has merit but does not fully meet PLOS ONE’s publication criteria as it currently stands. Therefore, we invite you to submit a revised version of the manuscript that addresses the points raised during the review process.

We look forward to receiving your revised manuscript.

Kind regards,

Lorenzo Faggioni, M.D., Ph.D.

Academic Editor

PLOS ONE

Reviewers' comments:

Reviewer's Responses to Questions

**Comments to the Author**

1. If the authors have adequately addressed your comments raised in a previous round of review and you feel that this manuscript is now acceptable for publication, you may indicate that here to bypass the “Comments to the Author” section, enter your conflict of interest statement in the “Confidential to Editor” section, and submit your "Accept" recommendation.

Reviewer #3: (No Response)

Reviewer #4: (No Response)

2. Is the manuscript technically sound, and do the data support the conclusions?

Reviewer #3: No

Reviewer #4: No

3. Has the statistical analysis been performed appropriately and rigorously? 

Reviewer #3: No

Reviewer #4: No

4. Have the authors made all data underlying the findings in their manuscript fully available?

Reviewer #3: No

Reviewer #4: Yes

5. Is the manuscript presented in an intelligible fashion and written in standard English?

Reviewer #3: Yes

Reviewer #4: Yes

6. Review Comments to the Author

Reviewer #3: ## Summary

- In this study, the authors focused on enhancing generalizability in classifying individual migraine patients and healthy controls using brain MRI data through a data harmonization strategy by using the Maximum Mean Discrepancy (MMD) in Geodesic Flow Kernel (GFK) space. Their results showed a homogeneous dataset derived from a cohort of healthy controls yielded a significant 25% accuracy improvement for both episodic and chronic migraineurs.

- The differences in participants characteristics between multi-institutional datasets are a significant issue and have already been discussed somewhere [1,2]. Since their methodology appears to be novel, and if the utility of this method is demonstrated, the contribution of this paper could be substantial. However, at present, the descriptions of how the methods and results were obtained are unclear, and the reliability of the results is low, which would make it difficult to justify publication of this paper. I have provided detailed points below.

## **Major Comments**

- **Unclear Description of Proposed Method**

- The description of the authors' proposed method is unclear, especially in Figure 1 where it is not evident what processes are being carried out. For instance, it is completely unclear how the 204 features are being utilized. It is also not clear whether MMD is calculated for each feature or feature vector. What does 'Multiple view' mean in the Figure 1?

- Looking at the algorithm in the Appendix, it seems like matching, but is it a method of extracting a similar group of subjects from both Dataset 1 and Dataset 2 in equal numbers?

- It is written, “To address this challenge, we propose a novel concept, the 'healthy core,' which involves transforming and aligning imaging data from HCs using their brain MRI data.” However, it is not clear if modulation applies to the features in the proposed method. If it does, how does it apply? There is no explanation for this.

- Moreover, with this algorithm, if there are identical subjects in both Dataset 1 and Dataset 2, it seems like it would end at i=1. How should the number of subjects be determined?

- **Unnatural and Unclear Aspects in Results**

- The results in Table 10 are labeled as accuracy, which I assume means, for example, in 'Predicting CM from DS2', how many out of 26 were correctly predicted. However, the numbers in the table do not correspond to any possible outcomes. For instance, for 'EM from DS2' with n=8, the accuracy should be one of 0.125, 0.25, 0.375, 0.50, 0.625, 0.75, 0.875, 1.0, but the numbers in Table 10 do not match any of these.

- Figures 3 and 4 appear unnatural to me. For example, even though the relative positions of the red dots for EM are exactly the same between Figures 4A and 4B, the scales of the x and y axes are different. This could be due to a mistake in the program and should be checked.

Also, in Figures 3 and 4CD, the scales differ too much between the left and right figures, and it seems like only the data for EM and CM are changing. This isn't just plotting EM and CM separately using t-SNE, right? It's applied to the integrated data, right? There is no explanation about the methodology in this regard, making it impossible to assess. Furthermore, since t-SNE depends on initial values and other factors, it might be necessary to also check with PCA or MDS.

- While some harmonization methods like Linear regression for noise removal and ComBat have been proposed [3], the authors' method is explained as being different from these methods. However, to demonstrate its utility, a direct comparison of the performance of the discriminators is necessary.

- The authors compare AUC=0.65 from 20 test data samples with AUC=0.62 from 70 test data samples between Experiment 1 and Experiment 2, but the difference in the number of samples in the Test is too large. Therefore, if the authors want to conclude from those results, some statistical test should be necessary.

## **Minor Comments**

- On page 12, the authors said that “The test results indicate that DS1 and DS2 are not compatible after mean and variance adjustments,” from which results can this be said?

- In the results of Table 4, the differences due to measurement bias and sampling bias are not distinguished (see [1]). If there is a significant difference between DS1 and DS2, could it be due to a large measurement bias?

- There is no description of how the hyperparameters for the machine learning methods were determined. What criteria were used on the validation dataset? AUC, accuracy, sensitivity, or specificity?

- Is the regularized logistic regression L1, L2, or elastic net? Given the clear overfitting of the results, L1 might be better.

- Tables 5 and 7 appear to be the same.

- Include validation results in Tables 6A and 6B as well.

- Not only AUC but also accuracy, specificity, and sensitivity should be displayed for discrimination performance. While I acknowledge the usefulness of AUC for unbalanced data, relying on a specific metric for evaluating the performance of a discriminator is risky, and it is important to consider these metrics holistically.

- On page 14, it states, “This deterioration in accuracy is expected.” Why is this anticipated? If the model is designed to avoid overfitting, deterioration should not occur. Is this statement necessary?

- "In Experiment 3, when the authors mention 'without healthy core', does this mean the authors examined the sensitivity as in Experiment 2? In other words, training and validation were following Table 7, and then using only CM and EM for the test?”

[1] https://sci-hub.se/downloads/2021-08-31/6f/maikusa2021.pdf

[2] https://journals.plos.org/plosbiology/article?id=10.1371/journal.pbio.3000042

[3] Johnson W E, et al. (2007) Adjusting batch effects in microarray expression data using empirical Bayes methods. Biostatistics, 8, 118–127.

Reviewer #4: In this study, the authors introduce a data harmonization approach, utilizing Maximum Mean Discrepancy, to identify a homogeneous healthy core from multicenter studies. They apply this method to two datasets with both healthy controls and migraine patients.

The work carried out by the group could be interesting. However, the method description is challenging to read and many details are reported in the results. Statistical analyses heavily focus on AUC but do not assess the accuracy of the analyses (balanced accuracy, considering groups of different sizes, sensitivity, specificity, confidence interval). The validation employs groups of 3 subjects, making result generalization very difficult. To highlight the method's strengths, it would have been more useful to analyze controls versus the entire migraine sample without dividing between chronic and episodic cases. These issues unfortunately present major challenges to drawing meaningful conclusions from the data.

Major:

- In the introduction, it should be specified that the focus is on harmonizing already collected data (on extracted features) and not during the acquisition phase (sequence parameters). Some references are missing.

- In FreeSurfer, despite being defined as Talairach transformation, it is applied between the original volume and the MNI305 atlas. Correct? The atlas used for the parcellation of the 68 regions is missing. Why are subcortical regions not considered?

- Many method sections are reported in the results.

- Tests (e.g., BDI, STAI) conducted on patients are mentioned but not used in the analyses.

- Tables lack details in both demographics and statistics (accuracy, sensitivity, specificity, confidence interval) and should be grouped. Statistical demographic differences between controls of the two groups are not reported.

- Resampling methods (e.g., bootstrap), a statistical procedure that resamples a single dataset to create many simulated samples, were not used to enhance result generalization.

- It's unclear which variables (thickness, area, surface) the results of various models are associated with.

7. PLOS authors have the option to publish the peer review history of their article (what does this mean?). If published, this will include your full peer review and any attached files.

Reviewer #3: **Yes: **Ayumu Yamashita

Reviewer #4: No

---

## [Decision Letter · Decision Letter 2]

16 May 2024

PONE-D-23-16414R2HealthyCore: Harmonizing Brain MRI for Supporting Multicenter Migraine Classification StudiesPLOS ONE

Dear Dr. Wu,

Thank you for submitting your manuscript to PLOS ONE. After careful consideration, we feel that it has merit but does not fully meet PLOS ONE’s publication criteria as it currently stands. Therefore, we invite you to submit a revised version of the manuscript that addresses the points raised during the review process.

We look forward to receiving your revised manuscript.

Kind regards,

Lorenzo Faggioni, M.D., Ph.D.

Academic Editor

PLOS ONE

**Additional Editor Comments:**

I commend the authors on considerably improving their manuscript over the original submission. However, there are still some issues that should be addressed as per the reviewer's comments, with the main goal to put the study findings into perspective and strengthen the discussion section.

Reviewers' comments:

Reviewer's Responses to Questions

**Comments to the Author**

1. If the authors have adequately addressed your comments raised in a previous round of review and you feel that this manuscript is now acceptable for publication, you may indicate that here to bypass the “Comments to the Author” section, enter your conflict of interest statement in the “Confidential to Editor” section, and submit your "Accept" recommendation.

Reviewer #4: (No Response)

2. Is the manuscript technically sound, and do the data support the conclusions?

Reviewer #4: Partly

3. Has the statistical analysis been performed appropriately and rigorously? 

Reviewer #4: Yes

4. Have the authors made all data underlying the findings in their manuscript fully available?

Reviewer #4: Yes

5. Is the manuscript presented in an intelligible fashion and written in standard English?

Reviewer #4: Yes

6. Review Comments to the Author

Reviewer #4: The use of a ‘healthy core’ can be helpful in multicentre studies. I appreciate the changes made to the text, however, there are still points that require further review.

1. The discussion lacks a comparison with previous works and references. The only sentence where works are mentioned is 'Many prior publications (including some of our own) reported the accuracy of classification models for migraine train and test using the same dataset, with strategies such as the "leave-one-out" methodology [Schwedt, T.J. et al. 2015, Chong, C.D. et al., 2017, Lee M.J. et al., 2019, Zhang, Q. et al., 2018, Yang, H. et al., 2018, Chong, C.D., 2021].' It is not possible to evaluate the goodness of the method if not compared to other models, even those not on migraines.

2. There are too many tables and they are not well structured:

Combine Table 1 and Table 2 together. Follow the example of Schwedt 2015. Insert variables age and gender in separate rows.

Combine Table 6, 8, and 10 indicating with a row the experiment (1, 2, 3) they are linked to.

The tables lack explanatory captions. For example, what the symbol > or arrow (e.g., CM1>CM2) means.

Combine Table 7A and 7B together. Same for Table 9A and 9B.

Why was a different terminology used in Table 11 than in Table 10? Predicting CM from DS2 instead of DS1 arrow DS2? Choose the same terminology in all tables for consistency.

3. Reformulate the sentence related to Freesurfer's recon-all. The message is incorrect. The recon-all allows obtaining cortical surfaces. By applying the Desikan atlas, it is possible to obtain volume, area, and thickness measurements for cortical regions.

4. Check the acronyms. For example, at the beginning of the results, acronyms already used that had not been defined such as EM and CM are reported.

5. There are several sentences in the results that should go in the discussion. E.g., page 18: ‘These significant enhancements … improvements across all evaluated performance metrics’.

6. Modify the title of the paragraph '3.4. Discussions on Healthy Core'. It cannot be defined as a 'discussion'. It seems more like an in-depth analysis. I emphasize that this part of the result is not reported in the method section.

7. PLOS authors have the option to publish the peer review history of their article (what does this mean?). If published, this will include your full peer review and any attached files.

Reviewer #4: No

---

## [Author Response · Author response to Decision Letter 2]

25 Jun 2024

We want to thank to the editor and the reviewer for the insightful review. We have carefully addressed all the comments and suggestions provided by the reviewer and believe that these revisions have significantly improved the quality and clarity of our manuscript.

In this submission, we have included a detailed point-by-point response to each comment raised by the reviewer in the response letter along with the corresponding changes highlighted in the revised manuscript.

We believe that our proposed 'healthy core' methodology offers a novel and robust approach to harmonizing multicenter brain MRI data and improving the generalizability and accuracy of migraine classification models. We hope that the reviewers and readers will find our contributions valuable.

---

## [Decision Letter · Decision Letter 3]

16 Jul 2024

HealthyCore: Harmonizing Brain MRI for Supporting Multicenter Migraine Classification Studies

PONE-D-23-16414R3

Dear Dr. Wu,

We’re pleased to inform you that your manuscript has been judged scientifically suitable for publication and will be formally accepted for publication once it meets all outstanding technical requirements.

Kind regards,

Lorenzo Faggioni, M.D., Ph.D.

Academic Editor

PLOS ONE

Reviewers' comments:

Reviewer's Responses to Questions

**Comments to the Author**

1. If the authors have adequately addressed your comments raised in a previous round of review and you feel that this manuscript is now acceptable for publication, you may indicate that here to bypass the “Comments to the Author” section, enter your conflict of interest statement in the “Confidential to Editor” section, and submit your "Accept" recommendation.

Reviewer #4: All comments have been addressed

2. Is the manuscript technically sound, and do the data support the conclusions?

Reviewer #4: Yes

3. Has the statistical analysis been performed appropriately and rigorously? 

Reviewer #4: Yes

4. Have the authors made all data underlying the findings in their manuscript fully available?

Reviewer #4: Yes

5. Is the manuscript presented in an intelligible fashion and written in standard English?

Reviewer #4: Yes

6. Review Comments to the Author

Reviewer #4: I have no further comments to add. the authors have attempted to respond to my comments. they improved the reading of the text

7. PLOS authors have the option to publish the peer review history of their article (what does this mean?). If published, this will include your full peer review and any attached files.

Reviewer #4: No

---

## [Editor Report · Acceptance letter]

24 Jul 2024

PONE-D-23-16414R3 

PLOS ONE

Dear Dr. Wu, 

I'm pleased to inform you that your manuscript has been deemed suitable for publication in PLOS ONE. Congratulations! Your manuscript is now being handed over to our production team.

Kind regards, 

on behalf of

Dr. Lorenzo Faggioni 

Academic Editor

PLOS ONE